# Low Energy Implantation of Carbon into Elastic Polyurethane

**Ilya A. Morozov [1],\* , Alexander S. Kamenetskikh [2], Anton Y. Beliaev [1], Marina G. Scherban [3] and Dmitriy M. Kiselkov [4]**

[1] Laboratory of Micromechanics of Media with Inhomogeneous Structure, Institute of Continuous Media Mechanics UB RAS, Academika Koroleva st., 1, Perm 614013, Russia; belyaev@icmm.ru
[2] Charged-Particle Beams Laboratory, Institute of Electrophysics UB RAS, Amundsen st., 106, Ekaterinburg 620016, Russia; alx@iep.uran.ru
[3] Faculty of physical chemistry, Perm State University, Bukireva st. 15, Perm 614990, Russia; ma-sher74@mail.ru
[4] Laboratory of Multiphase Dispersive Systems, Institute of Technical Chemistry UB RAS, Academika Koroleva st., 3, Perm 614013, Russia; dkiselkov@yandex.ru
\* Correspondence: ilya.morozov@gmail.com; Tel.: +7-342-237-8315

**Abstract:** Ion modification of polymeric materials requires gentle regimens and subsequent investigation of mechanical and deformation behavior of the surfaces. Polyurethane is a synthetic block copolymer: A fibrillar hard phase is inhomogeneoulsy distributed in a matrix of soft phase. Implantation of carbon ions into this polymer by deep oscillation magnetron sputtering (energy—0.1–1 keV and dose of ions—$10^{14}$–$10^{15}$ ion/cm$^2$) forms graphene-like nanolayer and causes heterogeneous changes in structural and mechanical properties of the surface: Topography, elastic modulus and depth of implantation for the hard/soft phase areas are different. As a result, after certain treatment regimens strain-induced defects (nanocracks in the areas of the modified soft phase, or folds in the hard phase) appear on the surfaces of stretched materials. Treated surfaces have increased hydrophobicity and free surface energy, and in some cases show good deformability without any defects.

**Keywords:** ion implantation; polyurethane; stiffness; deformability

## 1. Introduction

Materials with carbon or carbon-containing plasma-modified surface have found wide application in products with improved biomedical properties [1,2]. The treatment of soft polymers is of great interest. In this case, close attention should be paid to the mechanical and deformation properties of the coating. Regardless the nature of treatment, the stiffness of the coating (or the modified surface layer), even at the initial stage of formation [3], quickly becomes significantly higher than the polymeric substrate. A mechanical loading damages such surfaces, that is widely shown for different polymers and techniques of treatment: Carbon sputtering [4], cold plasma [5], plasma enhanced chemical vapor deposition [6], plasma ion implantation [7], etc. In a case of multi-cycle deformation, which is typical for real operating conditions, the cracks propagate to a depth exceeding the thickness of the modified layer, significantly damaging the substrate [8]. Whether the use of such materials is safe or not is an open question. Obvious, that the damage of the surface should be avoided.

A low-energy ion implantation could be the solution. In this case, ions will be distributed both on the surface and in some surface layer. An appropriate dose of ions will provide low stiffness, i.e., high deformability of the modified surface. In comparison with sputtering, changes in the structure of the

polymer (appearance of free radicals and hydrogen bonds) caused by the ion implantation will have a positive effect on the properties (high hydrophobicity and free surface energy) that are important for biomedical materials.

Polyurethane is a widely used synthetic polymer. Depending on a formulation, its mechanical properties vary from viscous liquids to rigid plastics. In particular, elastic polyurethanes are used in the manufacture of biomedical products (tubes, implants, etc.). Polyurethane has a complex chemical structure. It is a block polymer with domains of hard phase in a matrix of soft phase [9,10]. The study of changes of such polymer at an initial stage of ion treatment is the key for the development of the materials with good deformability. Note, that prolonged treatment creates homogeneous and undesirable hard surface.

The purpose of this work was to study the effect of low-energy carbon ion implantation on the structural and mechanical properties of elastic polyurethane, the formulating conditions of creating deformable and defect-free surfaces.

## 2. Materials and Methods

### 2.1. Creating Polyurethane

Polyurethane (PU) was manufactured from prepolymer (urethane based simple polyester and toluene diisocyanate) and cross-linking agent (hardener MOCA—16.5 weight parts and solvent polyfurite (polyoxytetramethylene glycol)). The components were heated up to 70 °C, vacuumed for 5 min, mixed, then vacuumed again and cured in a mold with an open top at 100 °C for 20 h. The thickness of the obtained plates was 2 mm. Initial elastic modulus measured by uniaxial mechanical tests—25 MPa, and elongation at break—800%. This PU consists of hard and soft blocks. Its chemical formula is shown in Figure 1.

**Figure 1.** Chemical structure of the polyurethane. Each hard block accompanied by *n* = 36 soft blocks.

### 2.2. Carbon Implantation

A flat balanced magnetron with a graphite target diameter of 80 mm was used as a source of ions, which worked in high-current modulation of pulsing current discharge (deep oscillation magnetron sputtering, DOMS). The ions coming from the plasma of magnetron discharge are accelerated near the sample by a metal mesh with grid size of 1.2 by 1.2 mm. The potential *U* of the mesh was set to 0.1, 0.3, 0.5 and 1.0 keV.

The samples were placed in a vacuum chamber on a water-cooled holder. The chamber was pumped to the pressure $5 \times 10^{-5}$ Torr. The partial pressure of argon was set to $2 \times 10^{-3}$ Torr and the surface of the magnetron target was cleaned in plasma of its own discharge for 5 min. During the cleaning, a shield was installed between the magnetron and the sample.

The treatment was carried out continuously in a pulsed mode: The amplitude of current discharge—40 A, pulse duration—8 μs, time between pulses—10 ms. The surface temperature of the samples was controlled by infrared pyrometer and did not exceed 30 °C.

The number *N* of pulses of magnetron discharge was 100 or 250. The average dose of ions per pulse was $6 \times 10^{12}$ ions/cm$^2$, i.e., the total dose was $6 \times 10^{14}$ ions/cm$^2$ or $1.5 \times 10^{15}$ ions/cm$^2$.

Choice of treatment parameters is based on the prevention of surface heating. In addition, preliminary studies have shown that higher energy or dose of ions produces stiff wrinkled surface that does not meet the deformability requirements.

### 2.3. Atomic Force Microscopy

An atomic force microscope (AFM) Ntegra Prima (NT-MDT B.V., Moscow, Russia) was used in a regime of dynamic nanoindentation: The probe indents a surface with high frequency (~1 kHz). Each point of the relief has its own dependence of load vs. depth of indentation. The indentation load was limited to 1.5–2 nN, so that the depth of indentation did not exceed 10–15 nm. Subsequent data processing gives a map of surface elastic modulus using a Johnson–Kendall–Roberts model. Obviously, the obtained values for the ion-treated surface are not the true modulus of the modified layer, but can be used for comparative analysis of the results.

Probes with calibrated tip radius (5 nm) and cantilever stiffness (0.4 nN/nm) were used. The shapes of the probes were estimated by the blind reconstruction method based on images of a test sample (TipCheck). The obtained AFM-images were subjected to partial restoration of true relief [11].

The materials were studied in both undeformed and stretched state. In the latter case, samples (20 by 2 by 2 mm) were glued to the substrate. The middle part was stretched by placing inserts between the sample and substrate. As a result, the top central part of the sample is stretched to a deformation of ~50%.

### 2.4. Raman Spectroscopy

Raman spectra were obtained by Bruker Senterra spectrometer with 532-nm excitation laser: Duration of accumulation—1 s; number of accumulations—1200; slot aperture—25 by 1000 μm. The radiation power was limited to 0.2 mW to prevent destruction of the soft polymer and to collect data from the very top layer of the surface. The base line of the spectra was leveled by adaptive method in Spectragryph software v1.2. Four spectra were obtained from different parts of the sample; the presented results are the average of these measurements.

### 2.5. Free Surface Energy

A wetting contact angle was determined by a sessile drop method. Water and diethylene glycol were used as test fluids. Free surface energy was calculated by an Owens–Wendt–Rabel–Kaelble method as the sum of dispersion (takes into account Van der Waals interactions) and polar (dipole interactions and energy of hydrogen bonds) components.

## 3. Results and Discussion

The surface of the untreated PU has a complex hierarchical morphology with structural–mechanical heterogeneities. Protruding surface irregularities visible on the microscale (Figure 2a) are agglomerates of the hard phase of the polymer: Elastic modulus of these elevations is higher that the surrounding area. At submicron level (Figure 2b) the hard phase has a tangled fibrillar nanostructure. The concentration of fibrils in the agglomerates (area "B" in Figure 2b) is higher than the low-modulus lowlands (area "A"); the latter correspond to areas with high concentration of soft phase.

The distributions of elastic modulus in relatively stiff $E_{hard}$ (bright areas of the modulus map) and soft $E_{soft}$ (dark areas) areas are shown in the insert in Figure 2a. Modal values are 15 and 25 MPa for soft and hard phases, respectively.

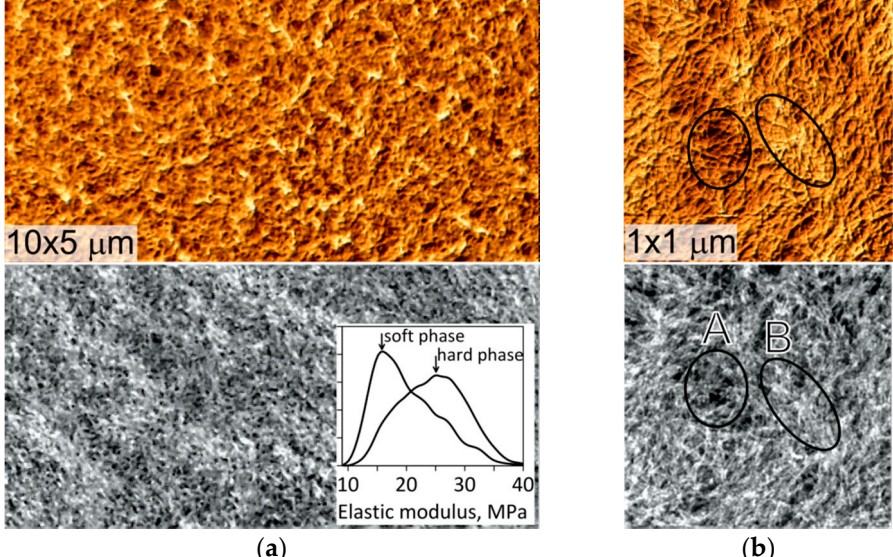

**Figure 2.** Atomic force microscope (AFM)-images of the relief (top row) and elastic modulus (bottom row) of the raw polyurethane surface on the scale (**a**) 10 by 5 μm and (**b**) 1 by 1 μm. An insert on (**a**) shows the distributions of the elastic modulus corresponding to areas of high concentration of soft (region "A" in Figure 2b) and hard (region "B") phases of the polyurethane.

The implantation depth of carbon ions was estimated using a Trim software [12]. The sizes of structural–mechanical inhomogeneities of the surface are from units (thickness of fibrils) to hundreds of nanometers (agglomerates). Therefore, it is reasonable to perform the calculations separately for the hard and soft phases (the chemical structure is presented in Figure 1). The results of the simulation (Figure 3) show that carbon ions penetrate deeper into the hard phase than into the soft phase. Therefore, the concentration of the implanted carbon in the unit volume of the soft phase is higher. This difference increases as the energy of ions rises.

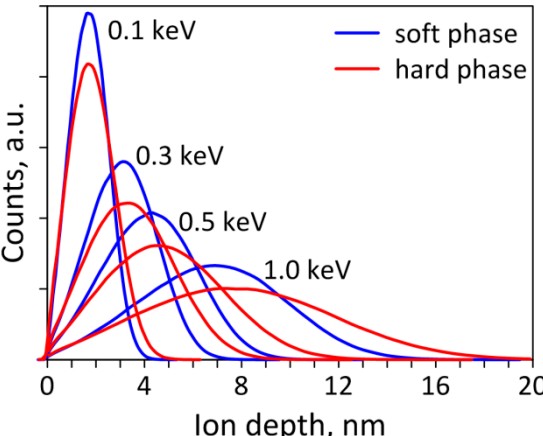

**Figure 3.** Simulation of the distribution of the depth of carbon ions implanted into the hard and soft phases of the polyurethane.

Representative AFM-images of treated surfaces are given in Figures 4 and 5. Elastic modulus shows clear contrast between the treated areas of hard (bright areas) and soft (dark areas on the modulus maps that correspond to the lowlands of the relief) phase.

Roughness of the modified soft phase of the PU rises as the energy of ions increases (Figure 6a). In these areas (highlighted by arrows in Figure 4), local wrinkles start to form at the lowest energy of ion implantation. When the energy of ions reaches 1.0 keV, the entire surface of the material is

covered with a chaotic wrinkled structure. The roughness of the modified hard phase does not change significantly up to the energy of 0.5 keV (Figure 6a), and then increases due to the wrinkling. Note that the wrinkles of the hard phase after 250 impulses of ion implantation are formed at 0.5 keV (see Figure 5c), but for $N = 100$ the higher energy is required (Figure 4d).

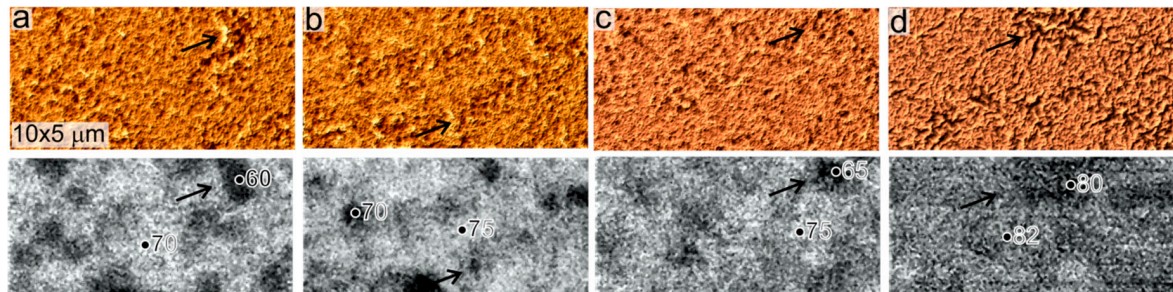

**Figure 4.** AFM-images of 10 by 5 μm surface reliefs (upper row) and elastic modulus maps (lower row) after ion implantation with 100 pulses and energy (**a**) 0.1, (**b**) 0.3, (**c**) 0.5 and (**d**) 1.0 keV. Hereinafter, each of the modulus maps is presented in its own scale. The markers indicate the values of modulus in MPa.

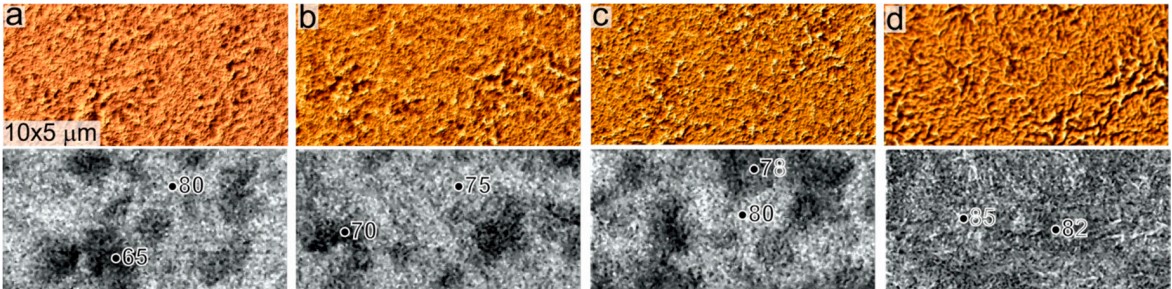

**Figure 5.** AFM-images of 10 by 5 μm surface reliefs (upper row) and elastic modulus maps (lower row) after ion implantation with 250 pulses and energy (**a**) 0.1, (**b**) 0.3, (**c**) 0.5 and (**d**) 1.0 keV. Hereinafter, each of the modulus maps is presented in its own scale. The markers indicate the values of modulus in MPa.

Wrinkling is caused by a loss of stability of the surface. A stress in a stiff layer on an elastic substrate at which the loss of stability occurs is proportional to the elastic modulus of the layer [13]. The values of $E_{hard}$ depend of the dose of ions and for the energy of 0.5 keV, 250 pulses are enough to wrinkle the hard phase. At higher energy of ions, $E_{hard}$ reaches one level regardless the ion dose (that is not the case for the modulus of modified soft phase), so the entire surface is covered with wrinkles. As for the modified soft phase, the rapid loss of stability can be explained by its properties (thickness and stiffness of the modified layer) and the influence of the surrounding hard phase.

Elastic moduli of the treated hard and soft phases increase from 60 to 90 MPa (Figure 6b). The increase of energy of ions leads to the homogenization of the mechanical properties of the surface: $E_{hard}/E_{soft}$ falls (see insert in Figure 6b). However, this trend is not satisfied for the materials with 100 impulses of ion implantation: At $U = 0.5$ keV a rise of mechanical heterogeneities of surfaces occurs. This slight jump of the properties will have a significant effect on the deformation behavior of the surface (see below).

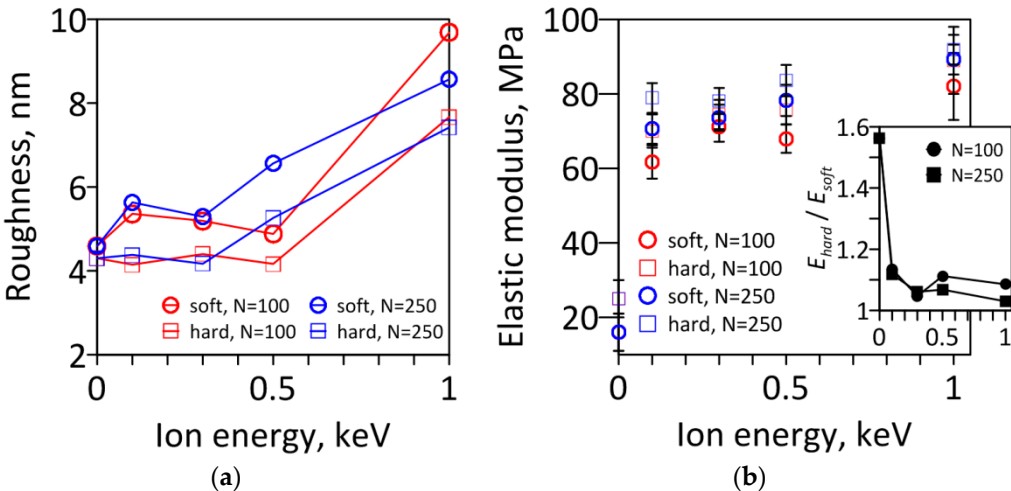

**Figure 6.** (**a**) roughness and (**b**) elastic modulus of the surfaces. The ratio of modulus of hard to soft phase is shown in insert.

Ion implantation breaks almost all fibrillar structures of the hard phase. Only separate filaments of the hard phase are yet visible if $U < 1.0$ keV (Figure 7a–c, Figure 8a–c). As energy of ions reaches 1 keV, the traces of the phase separation disappear (Figure 7d, Figure 8d). On the nano-scale (see inserts in Figures 7 and 8) at $U \leq 0.5$ keV, mesh-like nanostructure covers the surfaces. In the literature, one can find analogues of similar structures formed at the initial stage of carbon-like coating growth [14]. A height of cell walls is 0.4–0.6 nm, the size of cells is 10–20 nm. Qualitative analysis established the growth of mesh size as energy of ions increases.

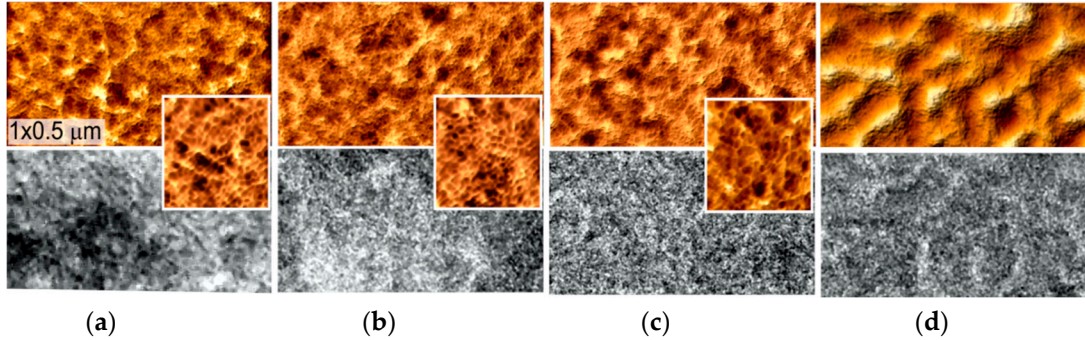

**Figure 7.** AFM-images 1 by 0.5 μm of surface micro-reliefs (upper row) and elastic modulus maps (lower row) after ion implantation with 100 impulses and energy (**a**) 0.1, (**b**) 0.3, (**c**) 0.5 and (**d**) 1.0 keV. The inserts are 200 by 200 nm.

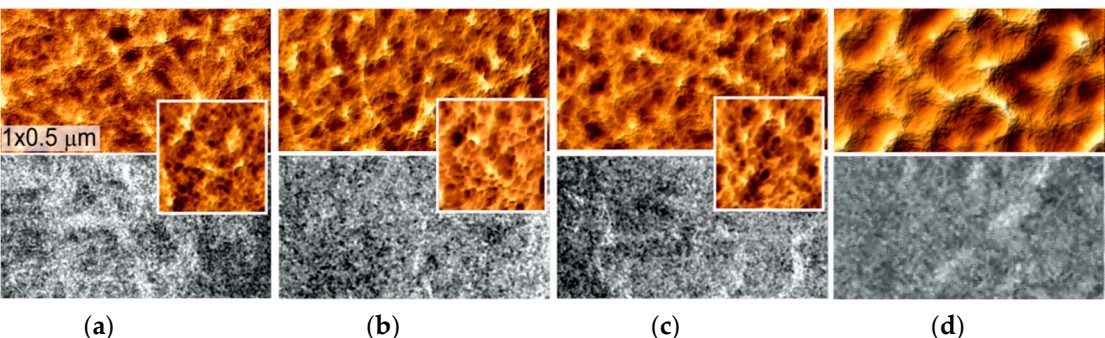

**Figure 8.** AFM-images 1 by 0.5 μm of surface micro-reliefs (upper row) and elastic modulus maps (lower row) after ion implantation with 250 impulses and energy (**a**) 0.1, (**b**) 0.3, (**c**) 0.5 and (**d**) 1.0 keV. The inserts are 200 by 200 nm.

In comparison with untreated polymer (Figure 2b), modulus maps of treated materials at nano-level do not correlate with the relief: Measurements are influenced by inhomogeneous mechanical properties of the modified surface layer of the material.

Figures 9 and 10 show AFM images of 50% stretched materials. The local stiffness of the surfaces has a determining influence on the mechanical behavior of the material during deformation.

The reliefs of stretched surfaces at $U \leq 0.3$ keV are similar: Carbon-modified agglomerates of the hard phase are oriented along the axis of deformation (Figure 9a); identical AFM images of such surfaces are not presented.

Submicron cracks are formed in soft areas of the stretched surface of the material treated with 0.5 keV and $N = 100$ (Figure 9b). The increase of energy to 1 keV leads to coalescence of cracks and increase of their width (Figure 9c), i.e., the fracture starts at lower critical tension. The average depth of cracks is 15–20 nm, that is more than the depth of ion implantation: High local strain (indicated by polymer strands connecting the crack edges, see Figure 9c) of the PU ruptures soft phase at the bottom of the cracks.

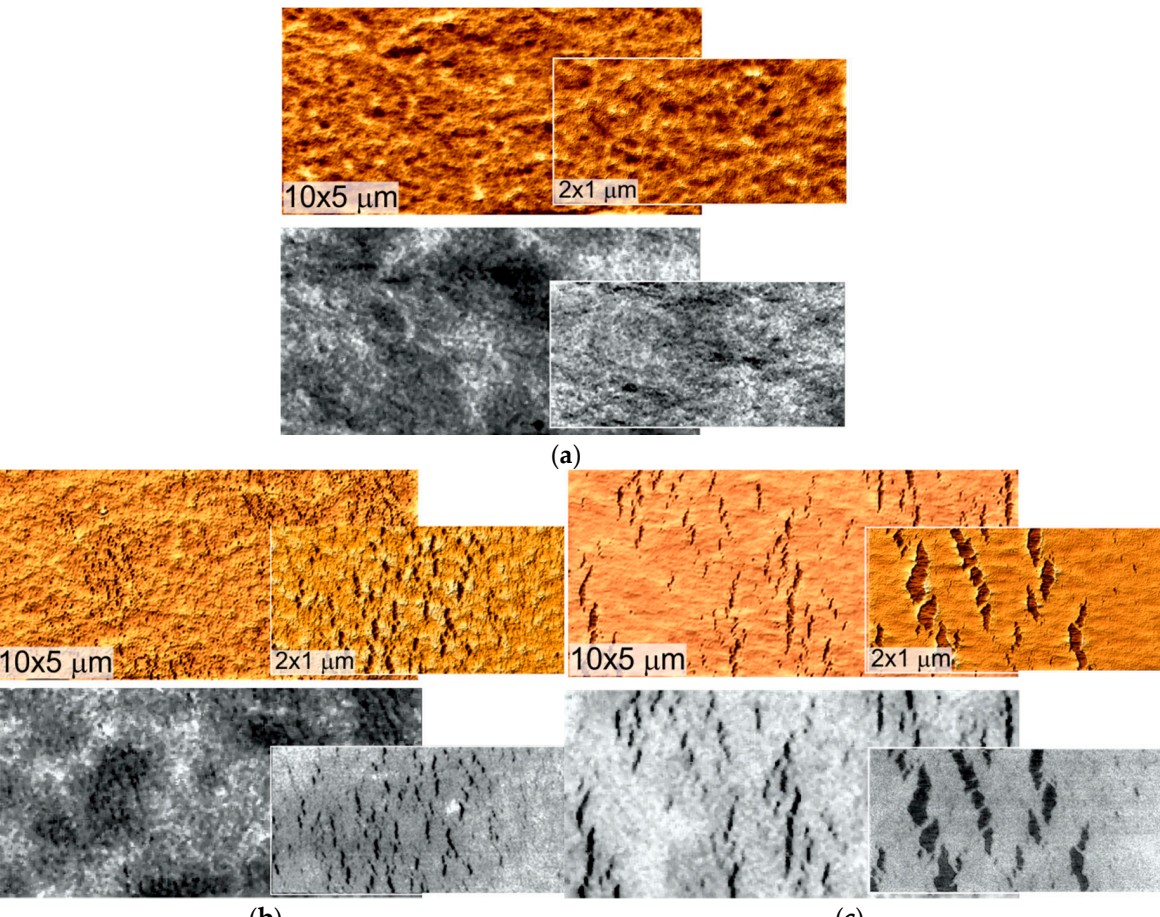

**Figure 9.** AFM-images 10 by 5 μm and 2 by 1 μm of 50% stretched materials after ion implantation with 100 pulses and energy (**a**) 0.3, (**b**) 0.5 and (**c**) 1.0 keV: Surface reliefs (upper row) and elastic modulus maps (lower row). The axis of elongation is horizontal.

The elastic modulus of a hard and soft phase after 250 impulses of implantation is higher (see Figure 6b) than for $N = 100$, however, damage of the surface does not occur (Figure 10): Strain-induced folds, co-directed with the axis of deformation appear first in the stiff areas (Figure 10a), and then expanding on the entire surface (Figure 10b): The elastic moduli of the modified phases reach one value. Note, that the wrinkles, caused by loss of stability, disappear under the tension.

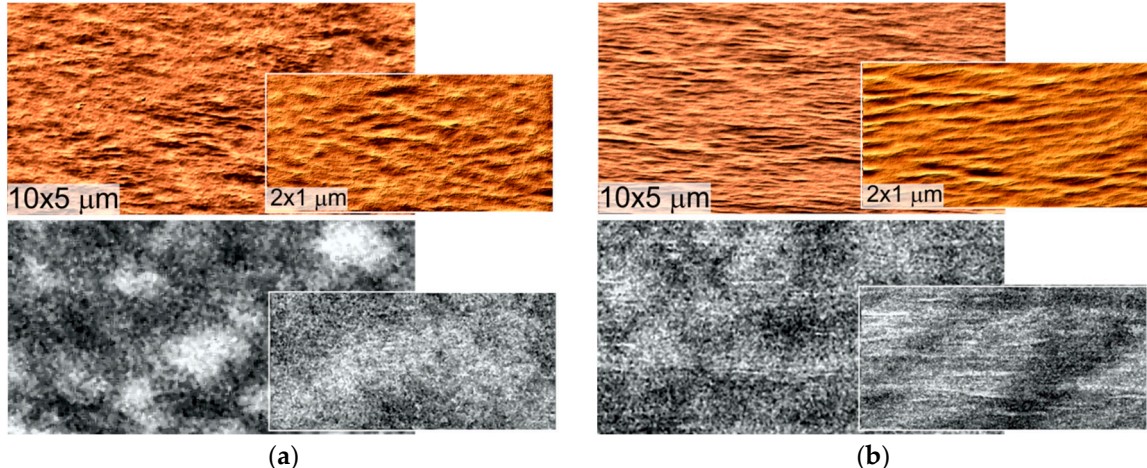

**Figure 10.** AFM-images 10 by 5 μm and 2 by 1 μm of 50% stretched materials after ion implantation with 250 pulses and energy (**a**) 0.5 and (**b**) 1.0 keV: Surface reliefs (upper row) and elastic modulus maps (lower row). The axis of elongation is horizontal.

The strain-induced folds (Figure 10) are quite sharp. These are the areas of potential crack nucleation and propagation under multicycle loads.

The deformation behavior is explained by the peculiarities of the elastic moduli of the treated hard and soft phase (see insert in Figure 6b): At 100 impulses of implantation the material is more favorable to break in less stiff and thinner modified layer of soft phase. At 250 pulses, the difference in stiffness is minimal, the stress field is more homogeneous, and, despite the higher stiffness of these surfaces, no damage occurs. Note, that no cracking was observed at $U = 0.1$ keV, when the difference in moduli is also significant. This can be explained by the small thickness of the modified layer (see Figure 3).

The internal stresses in the modified layer are nonlinearly dependent on the (low) energy of the implanted ions [15] and can be both tensile and compressive. The wrinkling in the areas of the modified soft phase is the sign of local compressive stresses. These internal stresses are amplified by influence of the surrounding hard phase. Thus, inhomogeneous stress field on the material surface layer could be another cause of cracking.

The treated materials have an increased free surface energy (Figure 11a). The growth is due to the polar component of the energy (characterizes dipole interactions and energy of hydrogen bonds), that should have a positive effect on the sorption activity of proteins [16]. Besides, low surface hydrophilicity is important for many biomedical applications (Figure 11b) [17]. It correlates with the polar energy component and maximal for $U = 1$ keV and $N = 100$, however, this material showed the worst deformation behavior.

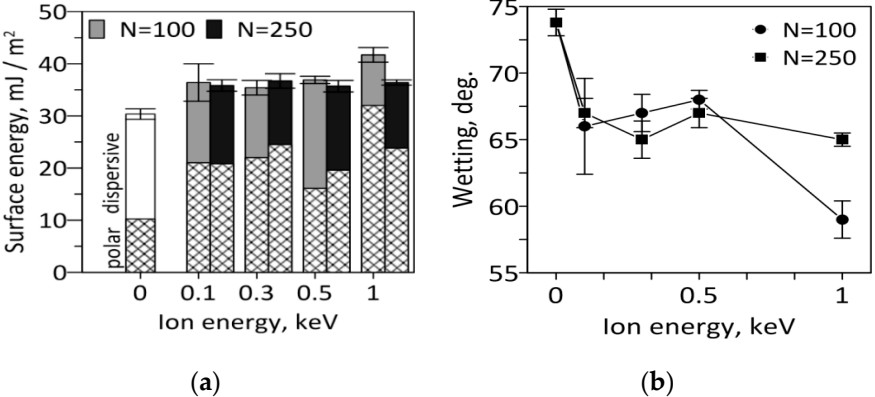

**Figure 11.** (**a**) free surface energy and (**b**) wetting contact angle.

The raise of the polar part is attributed to the increase of C=O groups and fall of dispersive part—reduction of carbonyl and benzene bonds (see further).

The destruction of polymer macromolecules leads to an emergence of free radicals. Theory predicts the formation of two types of free radicals [18]: Short-lived free radicals in damaged macromolecules and long-lived free radicals trapped in carbonized clusters. Carbonization of a polymer is usually achieved by high dose ($10^{16}$–$10^{17}$ ion/cm$^2$) of implanted plasma ions that recombine bonds between existing carbon atoms of the polymer macromolecules [19]. Such long-time treatment raises stiffness of the surface, making materials useless in deformation-related applications. In our case, carbon structures are formed by direct implantation of carbon, and the lower dose of ions creates softer surface.

Raman spectra of the materials are presented in Figure 12. The increased intensity in the region of 1800 cm$^{-1}$ corresponds to vibrations of C=O bonds; 2000–2200 cm$^{-1}$: C≡C, C=C bonds [20]. The intensity of carbonyl (1712 cm$^{-1}$) and benzene (1184, 1617 cm$^{-1}$) bonds of hard phase as well as N–C-N (1318 cm$^{-1}$) and C–O (1050 cm$^{-1}$) bonds decreases [21].

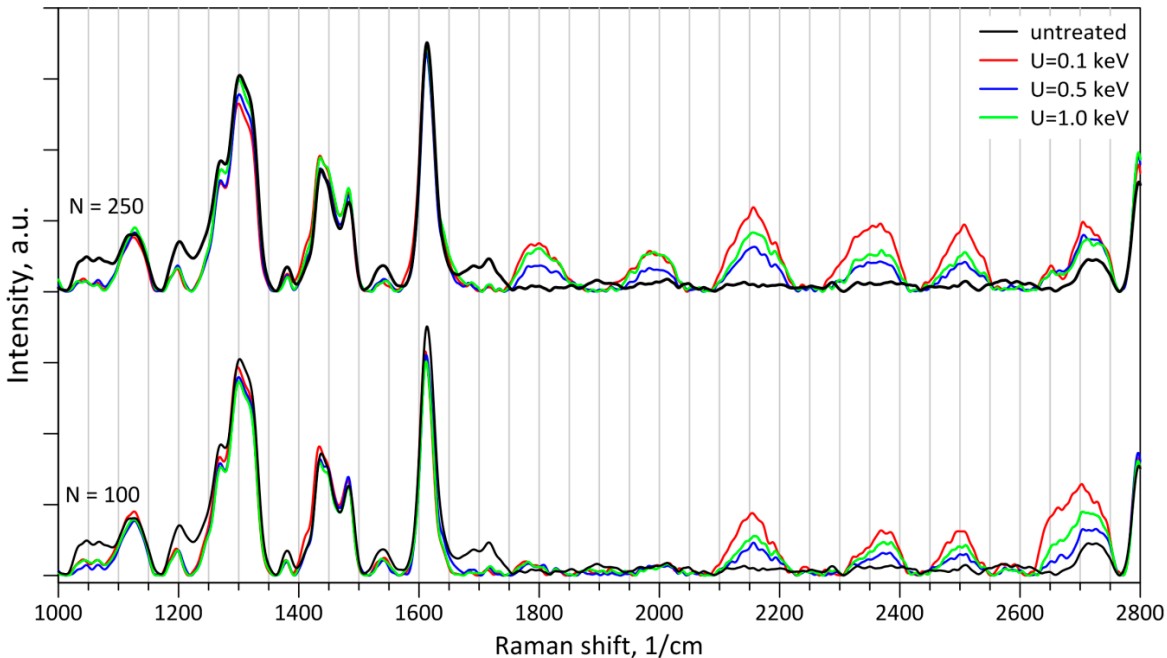

**Figure 12.** Raman spectra.

The peaks in the frequency range >2300 cm$^{-1}$ correspond to graphene-like structures [22–24]. The intensity of these peaks is higher at lower energy of ions. In this regime the carbon atoms have especially low implantation depth and, therefore, concentrate in a narrow surface layer forming single or stacking [24] graphene-containing structures: The monolayer of graphene [22] or disordered graphene [22] having sharp peaks at 2500 and 2700 cm$^{-1}$, graphene-oxide characterized by increased intensity at 2500–2700 cm$^{-1}$ [23], and the G-peak (~1600 cm$^{-1}$) of carbon is masked by the polyurethane. The mesh-like surface nanostructure (Figure 7a–c, Figure 8a–c) could be the direct evidence of such formations. Note, that more precise analysis is complicated due to heterogeneities of the original substrate and the modified surface layer. As the energy increases, the distribution of carbon ions over the depth broadens that decreases the intensity of graphene.

## 4. Conclusions

Low energy implantation of carbon ions into the elastic polyurethane surface forms graphene-like structures and causes heterogeneous structural and mechanical changes of the surface properties associated with the phase separation of the initial polyurethane into hard and soft blocks. Depending on the energy and dose of the ions, the local regions of the surface is modified not uniformly. Wrinkles

appear on the surface, first in the soft phase, then spreading over the whole surface. At the nano level, the fibrillar structure of the hard phase of the initial polymer is partially preserved up to the energy of 0.5 keV, and a mesh-like nanostructure appears on the surface. As the energy increases to 1.0 keV, the surfaces become homogenously stiff with a wrinkled relief.

The stiffness of the treated surfaces increases with the energy and dose of ions. However, the ratio of local elastic moduli of the modified hard and soft phases changes nonlinearly: At 100 pulses of implantation and energy of ions 0.5 or 1.0 keV the leap of mechanical inhomogeneities was observed. This is the reason of submicron size cracks in the soft areas of the surface of the stretched material. Treatment with 250 pulses leads to the equalization of the local mechanical properties of the surface and at energies of 0.5 and 1.0 keV no cracking occurs. However, sharp folds, which are aligned with the deformation axis, were observed on the stretched surfaces: Areas of potential defects under multi-cycle loads. Materials treated with an energy 0.3 keV and less, did not show any strain-induced defects.

Ion implantation decreases hydrophilicity and increases free surface energy, which makes these materials promising in developing flexible biomedical products. The optimal treatment regime for this polyurethane, in the terms of surface activity and deformability, can be considered as: Energy 0.3 keV and 250 impulses of implantation. In this case, an active and less hydrophilic surface with homogeneous stiffness is formed, deformable of at least to 50% without any defects.

**Author Contributions:** Conceptualization, Investigation, Writing—I.A.M.; Investigation, Methodology—A.S.K.; Investigation—A.Y.B., M.G.S. and D.M.K. All authors have read and agreed to the published version of the manuscript.

**Funding:** The work is supported by Russian Science Foundation, Grant 17-79-20042.

**Conflicts of Interest:** The authors declare no conflict of interest.

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
