# Peer review of "Low Energy Implantation of Carbon into Elastic Polyurethane"

_coatings, doi:10.3390/coatings10030274_

Round 1
Reviewer 1 Report
This is an interesting, well-organized paper that discusses the implantation of carbon ions into elastic polyurethane, paying special attention to the changes caused in this way in the structural, mechanical and physicochemical properties of the polymer surface. However, before the article will be accepted for publication, some minor and major points should be corrected:
– line 16 – superscripts should be corrected;
– Figure 1 – it seems to me that the chemical structure of polyurethane should be corrected. The chemical structure of toluene diisicyanate is questionable;
– line 149 – should be: (Figure 6a);
– line 236 – should be: Fig. 3;
– The statement that: "It is caused by the destruction of polymer macromolecules and emergence of free radicals, which should have a positive effect on the sorption activity of proteins..." (lines 244-245), is not clear. What radicals are formed? Are they stable in time? Is it not rather that the formed radicals react with water vapor and oxygen after contact with air of the treated polyurethane and thus form polar oxygen groups on the surface (e.g. -OH)? Could the Authors explain it and show what is really happening with the chemical structure on the surface?
– The contact angles shown in Fig. 11b are lower than 90 deg., so the hydrophobicity of the surfaces is rather poor in contrast to the statement in lines 245-246. Pleas arrange it.
– The discussion on contact angle measurements is very skimpy (lines 242-248). Could the Authors broaden this discussion and associate it with designated surface energy components? Which molecular structures are related to polar and dispersive components?
– Figure 12 is illegible. I propose to show one whole spectrum (e.g. for the untreated sample), and then only single selected and enlarged peaks that are discussed;
− How should graphene-like layer be understood (lines 267-268)? Please explain how the Authors imagine the molecular model of the polyurethane surface containing graphene structures.
Author Response
We thank the Reviewer for the time, spending with our manuscript and for the really interesting questions!
Major changes are highlighted in the revised manuscript.
– Figure 1 – it seems to me that the chemical structure of polyurethane should be corrected. The chemical structure of toluene diisicyanate is questionable;
The chemical structure was corrected.
– The statement that: "It is caused by the destruction of polymer macromolecules and emergence of free radicals, which should have a positive effect on the sorption activity of proteins..." (lines 244-245), is not clear. What radicals are formed? Are they stable in time? Is it not rather that the formed radicals react with water vapor and oxygen after contact with air of the treated polyurethane and thus form polar oxygen groups on the surface (e.g. -OH)? Could the Authors explain it and show what is really happening with the chemical structure on the surface?
First, we mention, that the cascade of carbon ions dramatically change the surface: each carbon ion has a probability to penetrate into material to a certain depth (see Fig. 3) changing the internal structure. On the way carbon ions can knock-out (and replace) the atoms of the polymer. These knocked atoms collide with neighbors and so on. Thus, the chemical structure of the surface (and modified nanolayer) is hard to imagine due to the complex structure of the polymer.
Ion implantation into polymers causes formation of two types of free radicals [18]: short-lived free radicals in damaged macromolecules and long-lived free radicals trapped in carbonized clusters. Carbonization of a polymer is usually achieved by high dose (1016…1017 ion/cm2) of implanted plasma ions that recombine bonds between existing carbon atoms of the polymer macromolecules [19]. Such long-time treatment raises stiffness of the surface, making materials useless in deformation-related applications. In our case, carbon structures are formed by direct implantation of carbon, and the lower dose of ions creates softer surface.
– The contact angles shown in Fig. 11b are lower than 90 deg., so the hydrophobicity of the surfaces is rather poor in contrast to the statement in lines 245-246. Pleas arrange it.
The discussion was changed to hydrophilicity.
– The discussion on contact angle measurements is very skimpy (lines 242-248). Could the Authors broaden this discussion and associate it with designated surface energy components? Which molecular structures are related to polar and dispersive components?
According to literature (Kondurin, Bilek. Ion Beam Treatment of Polymers) and our observations:
The raise of the polar part is attributed to the increase of carbon groups: C=O, C=C; fall of dispersive part – reduction of carbonyl and benzene bonds (see spectral analysis).
– Figure 12 is illegible. I propose to show one whole spectrum (e.g. for the untreated sample), and then only single selected and enlarged peaks that are discussed;
Improved
− How should graphene-like layer be understood (lines 267-268)? Please explain how the Authors imagine the molecular model of the polyurethane surface containing graphene structures.
The peaks in the frequency range > 2300 cm-1 correspond to graphene-like structures [22-24]. The intensity of these peaks is higher at lower energy of ions. In this regime the carbon atoms have especially low implantation depth and, therefore, concentrate in a narrow surface layer forming single or stacking [24] graphene-containing structures: the monolayer of graphene [22] or disordered graphene [22] having sharp peaks at 2500cm-1 and 2700 cm-1, graphene-oxide characterized by increased intensity at 2500…2700 cm-1 [23], and the G-peak (~1600 cm-1) of carbon is masked by the polyurethane. The mesh-like surface nanostructure (Figures 7a-c, 8a-c) could be the direct evidence of such formations. Note, that more precise analysis is complicated due to heterogeneities of the original substrate and the modified surface layer. As the energy increases, the distribution of carbon ions over the depth broadens that decreases the intensity of graphene.
Reviewer 2 Report
The topic is interesting with potential applications. The manuscript is well written, I only have some minor remarks below. I suggest the acceptance after the corrections.
Last sentence in the first paragraph of the introduction: Obviously, that > It is obvious, that
I suggest to write in the caption of Fig. 3 that it is a simulation.
For the AFM images a color bar with scale that shows the numerical height values is missing.
Second sentence in the conclusion: "modify not uniformly" > "is modified not uniformly"
Author Response
We thank the Reviewer for the time, spending with our manuscript!
> For the AFM images a color bar with scale that shows the numerical height values is missing.
The color bars are missing, because the height peculiarities are discussed in the text and reflected in roughness (Fig. 6a). As for the modulus maps, we added markers with values at extreme points. The average values and dispersions are reflected in Fig. 6b.
Therefore, we kindly ask the Reviewer not to insist about the colorbars. In our opinion they are almost uninformable and eating useful area.
Round 2
Reviewer 1 Report
Thanks the Authors for the response to my review.
The revised manuscript is better than its previous form, but it still contains some shortcomings that need clarification or improvement.
- The chemical formula has not been completely corrected.
As a result of the reaction between toluene diisocyanate and MOCA, you should get the following combination (yellow):
R1–N=C=O + R2–NH2 → R1–NH–(C=O)–NH–R2
In your chemical structure, the –NH– groups on the benzene rings of toluene diisocyanate are still lost in two places.
- First, we mention, that the cascade of carbon ions dramatically change the surface: each carbon ion has a probability to penetrate into material to a certain depth (see Fig. 3) changing the internal structure. On the way carbon ions can knock-out (and replace) the atoms of the polymer. These knocked atoms collide with neighbors and so on. Thus, the chemical structure of the surface (and modified nanolayer) is hard to imagine due to the complex structure of the polymer.
It is quite clear that the surface of the polymer that has been bombarded with carbon ions has a very complex structure. It seems, however, that no free radicals, but the polar groups (which you see as the polar component of the surface energy) have an effect on the sorption activity of proteins (page 246).
- Pages 250-251: The raise of the polar part is attributed to the increase of carbon groups: C=O, C≡C, C=C; fall of dispersive part – reduction of carbonyl and benzene bonds...
Are the C≡C and C=C groups polar? The polar part of surface energy increases with increasing concentration of polar groups such as C=O and C-OH (which can be seen by FTIR rather than Raman spectroscopy), but not nonpolar moieties such as C≡C or C=C. Please explain or correct it.
- Lines 262-263: The increased intensity in the region of 1800 cm-1 corresponds to vibrations of C=O bonds;
This is not visible in the new Figure 12.
- Figure 12
Once again, I suggest to show one whole spectrum (e.g. only for the untreated sample), and then only single selected and enlarged as much as possible peaks (for N=100 and N=250), which are discussed. Now both spectra are in practice the same (and, I guess, only for N=100, which is not marked). It would be much better for readers to prepare the figure as suggested.
Author Response
We thank Reviewer for the thoughtsprovoking remarks!
The formula in Fig. 1 was corrected.
We agree, that our conclusions about free radicals were overestimated.
These parts of the text were reworked.
We corrected the part about the polar groups and Figure 12 was reworked.
Kind regards,
Authors
Reviewer 2 Report
The suggested corrections have been made.
Author Response
Thank you for your time!